# Effect of Engine Design Parameters on the Climate Impact of Aircraft: A Case Study Based on Short-Medium Range Mission

Harjot Singh Saluja [1], Feijia Yin [1,*], Arvind Gangoli Rao [1] and Volker Grewe [1,2]

1 Faculty of Aerospace Engineering, Delft University of Technology, Kluyverweg 1,
2629 HS Delft, The Netherlands; h.s.saluja@tudelft.nl (H.S.S.); a.gangolirao@tudelft.nl (A.G.R.);
volker.grewe@dlr.de (V.G.)
2 Deutsches Zentrum für Luft- und Raumfahrt, Institut für Physik der Atmosphäre, Oberpfaffenhofen,
82234 Weßling, Germany
* Correspondence: f.yin@tudelft.nl

**Abstract:** The climate impact of aviation is considerably different from that of other transport modes. The turbofan engine's efficiency can be increased by increasing the Operating Pressure Ratio (OPR), bypass ratio (BPR) and Turbine Inlet Temperature (TIT), thereby reducing $CO_2$ and $H_2O$ emissions. However, this may have an adverse effect on the secondary emissions, such as $NO_x$, soot, etc. Taking a holistic view in evaluating the climate impact of engine development trends considering all the climate forcers is imperative for design trends in the future. This research investigates the impact of some key engine design parameters on climate. The emission changes due to design variations in the CFM56-5B are estimated using in-house engine performance and emission prediction tools. Accordingly, the changes in the species' Average Temperature Response for 100 years ($ATR_{100}$) are analyzed using a climate assessment tool, AirClim. The results show that the overall climate impact increases by 40% when increasing OPR from 25 to 40. Meanwhile, the Twin Annular Premixed Swirler (TAPS-II) combustor reduces the total $ATR_{100}$ drastically, in the range of 52–58%, due to lean combustion.

**Keywords:** turbofan engine design; aviation climate impact; aircraft emissions; combustion technology

## 1. Introduction

The impact of global aviation emissions in 2018 was estimated to be 3.5% of the total anthropogenic Effective Radiative Forcing (ERF) [1]. The aviation industry had been growing rapidly until the COVID-19 pandemic. The International Civil Aviation Organization's (ICAO) post-COVID-19 forecasts indicate that aviation will recover with an average growth rate of 3.6% until 2050 (https://www.icao.int/sustainability/Pages/Post-Covid-Forecasts-Scenarios.aspx (accessed on 28 September 2023)). As air traffic is expected to grow in the future, aviation's contribution to global climate change will be aggravated. Developing effective climate mitigation measures for the aviation sector is imperative.

Apart from the direct global warming effect from $CO_2$, a major contribution to aviation's climate impact is due to various non-$CO_2$ effects. These non-$CO_2$ effects include the direct water vapor ($H_2O$) emission effect [2], the indirect $NO_x$ emission effect via the ozone formation in the troposphere and the lower-most stratosphere, methane depletion [3,4], and the subsequent primary-mode ozone (PMO) effects, i.e., reduced ozone production due to the reduced $CH_4$ background concentrations [5], and the contrail effect [6–9]. Furthermore, non-volatile Particulate Matter (nvPM) from aircraft engines also has a greenhouse effect due to the absorption of outgoing longwave radiation [1]. In addition, nvPM also acts as ice nuclei, forming ice crystals under favorable atmospheric conditions, which affects the characteristics of contrails (e.g., lifetime and radiative effects), hence the actual climate impact of contrails [10]. Overall, these non-$CO_2$ effects contribute approximately two-thirds of the total aviation's net positive ERF [1]. While the impact of well-mixed greenhouse gases like

$CO_2$ is location-independent [11], the impact of non-$CO_2$ emissions depends on a variety of factors, such as the geographical location, altitude, time of emissions (e.g., day/night for contrails) and the local weather conditions [12,13]. Therefore, the local climate impact of aviation is not uniform across the earth, which needs to be considered for assessment.

Various efforts have been undertaken by organizations globally to address the challenge of the growing climate impact of aviation. Ambitious climate goals have been set up, which require rapid development and the adaptation of disruptive technologies. For instance, the Advisory Council for Aviation Research and Innovation in Europe (ACARE) has set the Flightpath 2050 targets to reduce the per passenger-km $CO_2$ emissions by 75% and the $NO_x$ emissions by 90% by 2050 relative to a typical new aircraft in the year 2000 [14]. The ICAO has introduced the Carbon Offsetting and Reduction Scheme for International Aviation (CORSIA), which aims to realize the carbon-neutral growth of the aviation sector. Grewe et al. (2021) [15] analyzed the climate impacts of different scenarios concerning measures like CORSIA and Flightpath 2050. By allowing aviation a 5% share of the goals of the Paris Agreement, they demonstrated that the Flightpath 2050 emission goals are in line with the target of limiting warming to 2 °C above pre-industrial levels [16], though they are not completely climate-neutral. In addition, it was also pointed out that with the currently expected technology enhancements, the emission goals are unlikely to be achieved.

However, simultaneously reducing $CO_2$ and $NO_x$ from the turbofan engine has been challenging because of the technological trade-off between fuel burn and $NO_x$ emissions [17–19]. One of the engine design trends has been to increase the Overall Pressure Ratio (OPR) combined with the increase in Turbine Inlet Temperature (TIT) to reduce the fuel burn, as can be seen in Figure 1, taken from Yin and Rao (2017) [18]. As stringent emission assessments were undertaken, it became apparent that the engine cycle development trend of pushing towards a high-pressure ratio resulted in a $NO_x$ penalty unless low $NO_x$ combustion techniques were applied [20,21]. Given such trade-offs, Freeman et al. (2018) [22] showed that to reach the "break-even" point at the overall RF, a 20% $NO_x$ reduction would only allow a 0.5% increase in $CO_2$. However, the actual sensitivity of $NO_x$ to $CO_2$ in a state-of-the-art aircraft system has been virtually left out of the discussion. Furthermore, despite the complex nature of the nvPM formation process, in general, the nvPM emissions increase along with the combustor inlet temperature [23]. This implies a trade-off between $CO_2$ and nvPM, and thus further trade-offs between the climate impact of $CO_2$ and non-$CO_2$ effects. Previous research proposed novel engine cycles to reduce the emissions of aircraft, and therefore the consequent environmental impact. Examples include Water-Enhanced Turbofans (WETs) [24] and composite engine cycle [25].

Apart from engine cycle development, combustion technology is considered significant in affecting $NO_x$ and nvPM emissions [26]. The state-of-the-art combustion technology in use can be categorized into rich-burn (represented by rich-burn, quick-quench, lean-burn (RQL) combustion technology) and lean burn (LB) (represented by the TAPS combustion technology [27] applied in GEnx and LEAP engines). Compared to RQL combustion, the TAPS combustion technology is expected to reduce both $NO_x$ and nvPM emissions substantially, which is beneficial from a climate impact point of view. Having a quantitative measure of such trade-offs in mind is essential for future engine design aspiring towards climate neutrality.

This paper aims to evaluate the climate impact of turbofan engine design in terms of cycle development and combustion technologies over time, with the Average Temperature Response over 100 years ($ATR_{100}$) being the climate metric. The analysis is carried out by varying two fundamental parameters for turbofan engines: the OPR and the TIT. Therefore, depending on these two parameters, the variation of fuel consumption, $NO_x$ emissions, nvPM particle number, and climate response in $ATR_{100}$ are examined in detail. Towards this end, we applied a complete model chain including an aircraft and engine performance model, emission models for $NO_x$ and nvPM prediction, and a state-of-the-art climate assessment model, AirCilm [5,28]. Section 2 of the paper elaborates on the methodology of

the research. Section 3 then presents the main findings of the research. Section 4 concludes the research findings and discusses further considerations of the analysis.

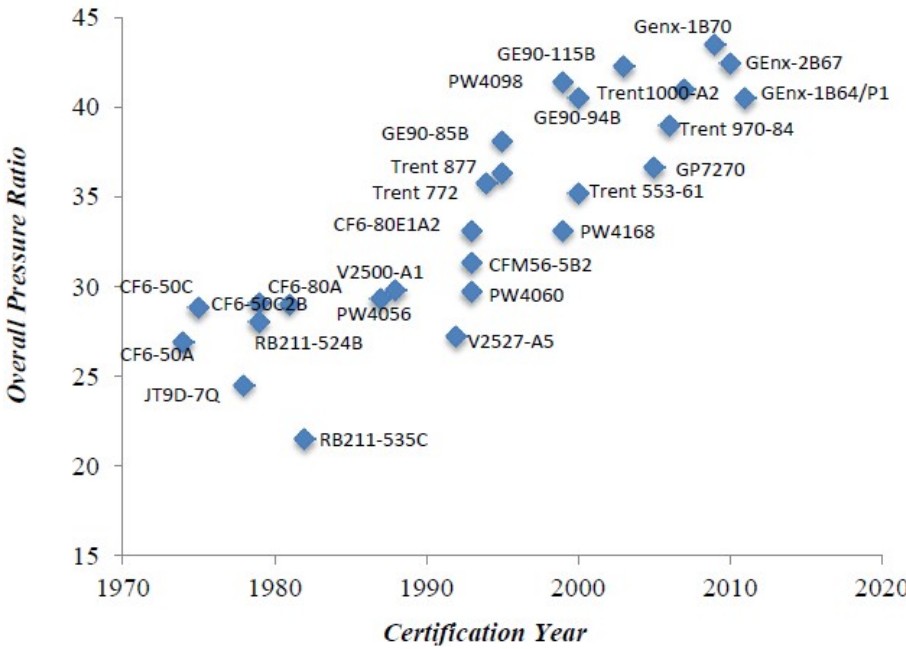

**Figure 1.** An overview of engine pressure ratio development over the years [18] (reprinted with permission).

## 2. Methodology

Figure 2 shows an overview of the model chain for this research. The methodology relies on a complete model chain consisting of aircraft and engine performance models, emissions models developed in-house for $NO_x$ and nvPM predictions and based on the P3-T3 method [29], and a state-of-the-art climate assessment model, AirClim [5,28]. There are three aspects of the climate impact evaluation of aviation technologies: the technology itself, the flight planning in which the said technology is applied, and the related atmospheric response. To determine the performance of the considered technology in terms of fuel consumption and emissions, we consider the various models, as shown in the figure. The flight planning is represented by the emission inventory based on these performance parameters, and the atmospheric response given by AirClim. For the emission inventory, we consider a total of 60 city pairs where A320 flights were operational in North America and Europe. The geographical location data for these city pairs, in the form of longitude–latitude pair, were obtained from the OpenSky Network [30], which is used to calculate the cruise distance. The number of passengers was taken to be 150, with 90 kg each, including baggage weight, giving the total passenger payload weight as 13.5 tons. For each city pair, the aircraft performance model along with the engine model calculates the thrust requirement during cruising, with the cruise altitude, $h_{cruise}$, being fixed at 33,000 ft, and the Mach no. at 0.78. The station thermodynamic data from the engine model are used to calculate the $NO_x$ and nvPM emissions from the in-house models. The geographical location data, fuel consumption, emissions, flight altitude and the frequency of flights for all the city pairs combine to form the emission inventory. The emission inventory serves as an input to AirClim [5,28], which then calculates the $ATR_{100}$.

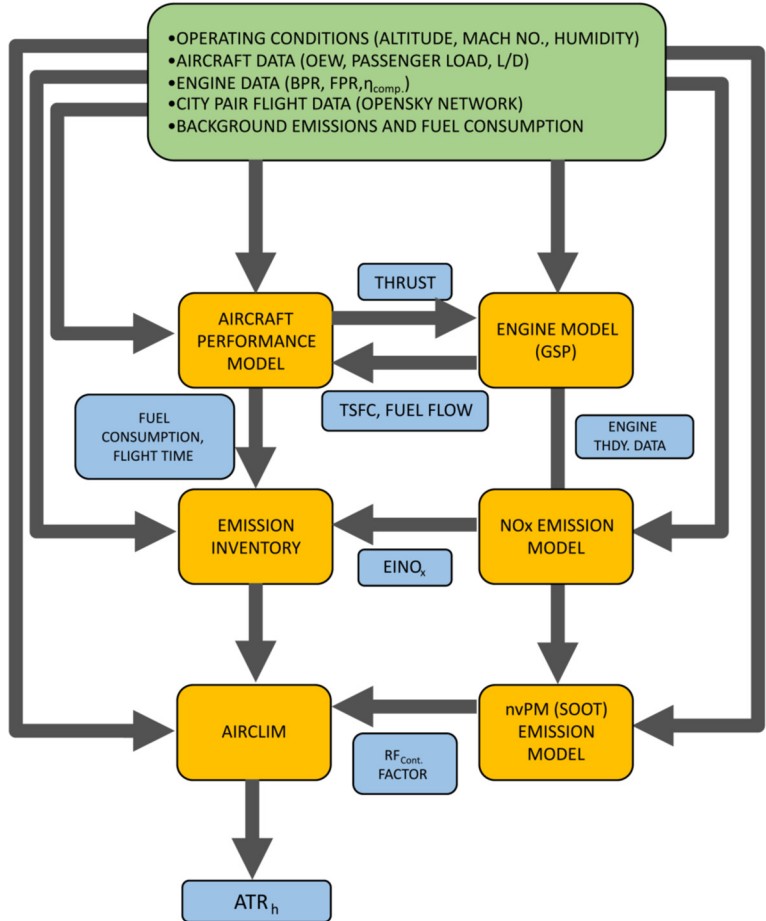

**Figure 2.** The overview of the modeling chain implemented in this research. The contents in green at the top represent the various boundary/operating conditions, those in yellow represent the various models, and those in blue represent the relevant parameters and outputs of interest.

## 2.1. Aircraft Performance Model

While there are aircraft performance tools available [31], for this research, a simple aircraft performance model was developed in-house, allowing better integration with the engine performance model, which is described in the subsequent section. The aircraft performance model uses the Breguet Range equation principles to estimate the cruise thrust requirement. As the aircraft weight reduces during cruising, the thrust requirement decreases at a constant Mach no. In this research, the average cruise thrust for the aircraft ($F_{avg,aircraft}$) was calculated using the simple force–balance equation for aircraft cruise [32], shown in Equations (1) and (2):

$$F_{avg.aircraft} = \frac{W_{avg.} \cdot g}{(L/D)} \tag{1}$$

where

$$W_{avg.} = \frac{W_{start} + W_{end}}{2} \tag{2}$$

$W_{start}$ is the aircraft weight at the beginning of cruising, consisting of the operating empty weight (OEW), the apparent payload weight ($W_{app.payload}$, composed of the payload weight and the total fuel for the descent, approach phases, and the reserves), and the required cruise fuel ($W_{f,cruise}$). $W_{end}$ is the aircraft weight at the cruise end, consisting only of the OEW and the $W_{app.payload}$. L/D is the lift-to-drag ratio. The $W_{app.payload}$ is obtained from the actual payload using a payload multiplication factor. With the average cruise

thrust and the cruise fuel consumption, the cruise time, $t_{cruise}$, could be calculated from the definition of thrust-specific fuel consumption (TSFC) [32], shown in Equation (3):

$$t_{cruise} = \frac{W_{f,cruise}}{TSFC \cdot F_{avg.aircraft}}$$

(3)

where TSFC is the thrust-specific fuel consumption. The cruise distance, $d_{cruise}$, could be determined using a simple distance–time relation for a given cruise speed.

The key model outputs, $F_{avg.engine}$, $t_{cruise}$ and $d_{cruise}$, obtained using this approach were compared with results from a commercial aircraft performance tool, PIANO-X (https://www.lissys.uk/PianoX.html (accessed on July 2023)). Three different aircraft were simulated in PIANO-X for their respective design range and standard payload conditions, as defined in PIANO-X. The details of these PIANO-X simulations are given in Appendix A.1. The comparison of the resultant aircraft performance parameters with those from PIANO-X is shown in Figure 3. Overall, the relative difference was ±2.5% with respect to PIANO-X results.

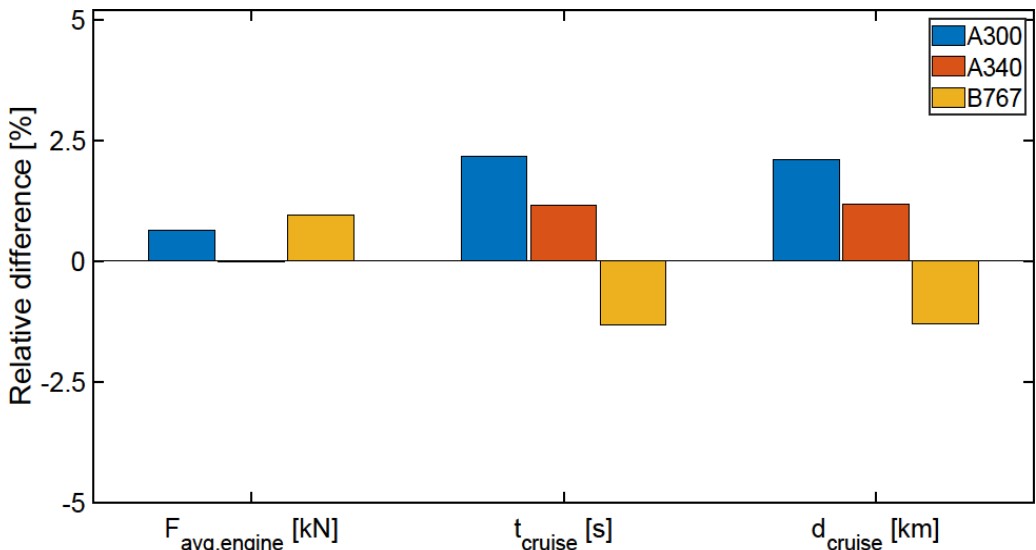

**Figure 3.** Relative difference of key aircraft performance parameters calculated by the in−house model with respect to PIANO−X output.

Further, PIANO was also used to determine the payload multiplication factor. Several simulations were carried out for different aircraft, and this factor was found to range from 1.17 to 1.25. In this research, this factor is assumed to be 1.22.

### 2.2. Engine Performance Model

In this research, engine modeling was carried out in the Gas Turbine Simulation Program (GSP) [33]. The engine model was coupled with the aircraft performance model, described previously, via the GSP API module, enabling automated engine performance calculations. The average engine cruise thrust, $F_{avg.engine}$, along with the $h_{cruise}$ and the Mach no., serve as inputs in GSP.

The CFM56-5B4/P (ICAO UID: 3CM026) was taken as the baseline engine model, for which the pressure ratio, the bypass ratio, and the fuel and emission characteristics for the landing and take-off (LTO) cycle were obtained from the ICAO engine emissions databank (https://www.easa.europa.eu/domains/environment/icao-aircraft-engine-emissions-databank (accessed on July 2023)).

Initially, GSP determines a design point and then calculates the output for off-design conditions. For this analysis, the design point was taken to be at the rated thrust setting at static sea-level (SLS) conditions. The intake air mass flow rate $\dot{w}_a$ was obtained as

377.75 kg/s to match the rated thrust under the SLS condition, and the TIT was found to be 1535 K. It should be noted that the design point does not refer to the condition for which the turbofan engine is designed; rather, design point is a reference operating condition where information about the engine parameters is known beforehand from the ICAO databank.

Once the engine was modeled under the design conditions, the cruise condition, i.e., the off-design condition, was simulated for each city pair. The resultant TSFC and cruise fuel flow rate $\dot{w}_{f,cruise}$ were then fed back to the performance model to determine the total cruise fuel consumption ($W_{f,cruise}$) and the $t_{cruise}$. Further, the engine's thermodynamic data from GSP were used in the emission models as input.

### 2.3. Emission Models

This section describes the emission models implemented in this study. The two main products of complete combustion, $CO_2$ and $H_2O$ are proportional to the fuel burn. These can be calculated using Equation (4) [34]:

$$W_{species,cruise} = \cdot EI_{species} \cdot W_{f,cruise} \tag{4}$$

where $W_{species,cruise}$ is the weight of the emission species in kg, and $EI_{species}$ is the emission index (EI) of the species in kg (species)/kg fuel. The EIs for $CO_2$ and $H_2O$ are taken to be 3.15 kg($CO_2$)/kg fuel and 1.25 kg($H_2O$)/kg fuel.

Estimating the non-$CO_2$ emissions of $NO_x$ and nvPM is not straightforward. These are largely dependent on the combustion parameters and power settings. For $NO_x$ emissions, the well-known P3-T3 method [29] was implemented, whereas for nvPM the approach was largely adapted from the method given by Durdina et al. (2017) [23]. In both methods, the EI is first calculated at a reference condition (in this study, the rated thrust setting under the SLS conditions), and then combustor inlet pressure ($P_{t3}$) and Fuel-to-Air ratio (FAR) corrections are applied to estimate the EI for a different operating condition (in this study, the cruise condition). Separately, in-house $NO_x$ and nvPM models, for the rich-burn and lean-burn combustion technologies, were developed to estimate the emissions at design conditions for the different engine configurations. These are subsequently described in more detail.

#### 2.3.1. Rich-Burn Combustor Emission Modeling

This section describes the $NO_x$ and nvPM emission models for the rich-burn RQL combustor.

#### Rich-Burn $EINO_x$ Model

For the cruise $EINO_x$ ($EINO_{x,cruise}$), the $P_3$–$T_3$ correlation is shown in Equation (5):

$$EINO_{x,cruise} = EINO_{x,SLS} \cdot \left(\frac{P_{t3,cruise}}{P_{t3,SLS}}\right)^n \cdot \left(\frac{FAR_{cruise}}{FAR_{SLS}}\right)^m \cdot \exp(h) \tag{5}$$

where the $EINO_x$ at the rated thrust, and under SLS conditions ($EINO_{x,SLS}$), is corrected due to the changes in $P_{t3}$, FAR and atmospheric humidity (h) at cruise conditions to obtain the $EINO_{x,cruise}$, with the same operating combustor inlet temperature ($T_{t3}$). $n$ and $m$ are the exponents of the pressure and FAR terms, respectively. To estimate the $^{EINO_{x,SLS}}$, the emission correlation was developed as a function of $P_{t3}$, $T_{t3}$ and FAR, as shown in Equation (6):

$$EINO_{x,SLS} = a \cdot P_{t3}^b \cdot \exp(c \cdot T_{t3}) \cdot d^{(f \cdot FAR)} \tag{6}$$

The ICAO LTO cycle data for three CFM56 models, simulated in GSP, were curved-fitted to obtain the constants in the model, as shown in Equation (7).

$$EINO_{x,SLS} = 0.1921 \cdot P_{t3}^{-0.7686} \cdot \exp(0.0084 \cdot T_{t3}) \cdot 2.01^{(60 \cdot FAR)} \tag{7}$$

This model was tested with two other CFM56 models (eight LTO cycle points) from the ICAO databank and was found to predict the EINO$_x$ within ±10% of the ICAO values, as shown in Figure 4.

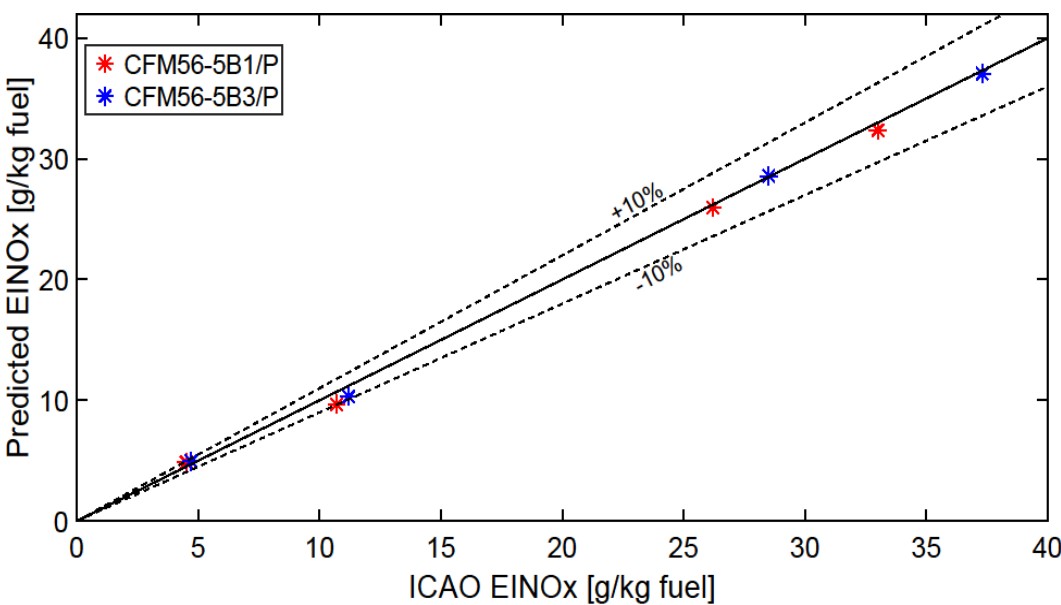

**Figure 4.** Comparison of the model−predicted EINO$_x$ (Equation (7)) with the ICAO−provided EINO$_x$ under SLS conditions for two CFM56 models (total eight points) for the rich−burn combustor. The solid dashed line is used as a reference to measure the accuracy of the model, with the dashed lines representing a ±10% deviation.

The values of *n* and *m* were taken to be 0.3 and 0, respectively, as suggested in the literature for a rich-burn combustion chamber [29]. The humidity correction was found to be approximately 1.13, taken from the ICAO Annex 16 for 60% relative humidity [29] (https://store.icao.int/en/annexes/annex-16 (accessed on July 2023)).

Rich-Burn nvPM Model

To estimate the nvPM particle number emissions during cruising, first the nvPM mass EI, $EI_{mass,cruise}$, was calculated, using the correlation from Döpelheuer and Lecht (1999) [35], as shown in Equation (8):

$$EI_{mass,cruise} = EI_{mass,SLS} \cdot \left( \frac{P_{t3,cruise}}{P_{t3,SLS}} \right)^{1.35} \cdot \left( \frac{FAR_{cruise}}{FAR_{SLS}} \right)^{2.5} \tag{8}$$

where $EI_{mass,SLS}$ is the nvPM mass EI under rated thrust–SLS conditions. The number emission index at cruise, $EI_{num,cruise}$, can be calculated using a parameter $\nu$, which is the ratio of $EI_{num,cruise}$ to $EI_{m,cruise}$, as shown in Equation (9). $\nu$ is a function of T$_{t3}$ and is independent of changes in the altitude [23].

$$EI_{num,cruise} = \nu \cdot EI_{mass,cruise} \tag{9}$$

The $EI_{mass,SLS}$ and $\nu$ were modeled as a function of T$_{t3}$ by curve-fitting nvPM data for six different CFM56 engines from the ICAO databank. These include corrections for system losses during measurements, as provided by the databank. $EI_{mass,SLS}$ was fitted with sixth-order polynomials, whereas $\nu$ was modeled as a decreasing exponential function of T$_{t3}$. These are shown in Equations (10)–(12), respectively.

$$EI_{mass,SLS} = (2.798 \cdot t^6) + (0.4 \cdot t^5) - (11.817 \cdot t^4) + (6.313 \cdot t^3) + (38.427 \cdot t^2) + (26.505 \cdot t) + 6.396 \tag{10}$$

$$\nu = (0.7944e + 14) \cdot \exp(-0.5956 \cdot t) \tag{11}$$

$$t = \frac{T_{t3} - 648.1}{139.2} \tag{12}$$

The $EI_{mass,SLS}$ correlation was tested with LTO cycle data for two other CFM56 engine models (eight LTO cycle points) from the ICAO databank and the model prediction was found to be within $\pm 5\%$ of the ICAO values for the take-off and climb-out thrust settings, as shown in Figure 5. For the lower thrust settings (approach and idle), the correlation became less accurate. As this correlation was to be typically used for cruise $T_{t3}$ values that correspond to thrust settings closer to the climb-out point, it was considered sufficiently accurate for this research.

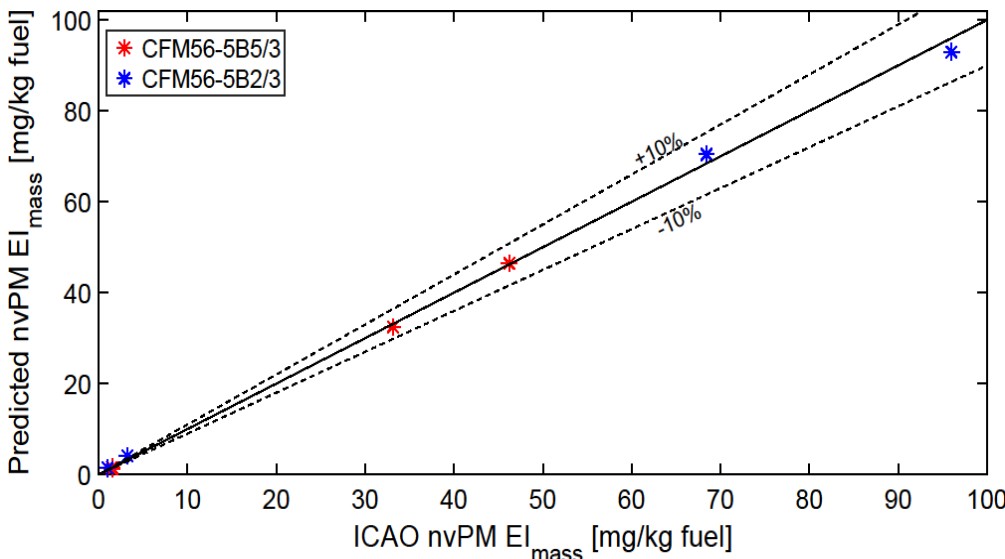

**Figure 5.** Comparison of the model−predicted nvPM $EI_{mass,SLS}$ (Equation (10)) with ICAO−provided nvPM under SLS conditions for two CFM56 models (total eight points) for the rich−burn combustor. The solid dashed line is used as a reference to measure the accuracy of the model, with the dashed lines representing a $\pm 10\%$ deviation.

Further, to test the validity of the model during cruise, it was applied to a test case given by Durdina et al. (2017) [23] for a Boeing 737NG equipped with a CFM56-7B at cruise. The engine was modeled in GSP, and the given cruise conditions were simulated to obtain the $T_{t3}$ for the subsequent calculation of nvPM EIs. It was found that the nvPM $EI_{mass,cruise}$ was predicted within $\pm 10\%$, and nvPM $EI_{num.,cruise}$ within $\pm 6\%$ using the model.

### 2.3.2. Lean-Burn TAPS-II Emissions Modeling

The TAPS combustion technology implemented in the next-generation LEAP-X engine has a lower $EINO_x$ due to its lean-burn operation. Foust et al. (2012) [36] described the development of the TAPS combustor and gave a comparison of the broad $NO_x$ emission trends of the TAPS combustor with those of the RQL combustor during typical aircraft operating points. For each fuel injector in a TAPS combustor, there is a pilot injector and a concentric outer main injector. The pilot has a rich-burn configuration, which is operational during low-power settings. At high power settings, the fuel is split between the pilot and the main, with most of the fuel injection through the main. The main injection is a set of radial jets that enter a larger air swirler, with a large effective area leading to a leaner fuel burn. The resultant lean combustion leads to lower $NO_x$ emissions. In this research, the $NO_x$ emissions for the TAPS combustor are modeled based on the trends given in [36]. As the thrust value for achieving such reduction has not been explicitly mentioned in the

public domain, in this research, it was assumed that the main burner begins to operate at 40% of the rated thrust for the engine.

Lean-Burn TAPS-II EINO$_x$ Model

Two separate correlations were developed for the pilot and pilot + main operations. ICAO LTO cycle data for six different LEAP engines were considered; that is, twelve LTO cycle data points in each operating regime. The format for these correlations is the same as that in Equation (6), and the model is shown in Equation (13).

$$\begin{aligned} EINO^p_{x,SLS} &= 0.0069 \cdot P_{t3}^{-2.7496} \cdot \exp(0.0216 \cdot T_{t3}) \cdot 9.773^{(13.3701 \cdot FAR)} & F_{engine} &\leq 40\% \\ EINO^{p+m}_{x,SLS} &= (3.9072 \cdot 10^{-6}) \cdot P_{t3}^{-13.9974} \cdot \exp(0.0776 \cdot T_{t3}) \cdot 1.4697^{(86.8456 \cdot FAR)} & F_{engine} &> 40\% \end{aligned} \qquad (13)$$

The above model was tested with LTO cycle data for three other LEAP engines (six data points for each correlation), and the accuracy of the prediction is shown in Figure 6. Specifically, the correlation for pilot operation showed greater accuracy. For the pilot + main operation, the prediction showed a slightly larger deviation. It is expected that as data for more LEAP engine models are provided in the ICAO databank in the future, the correlations could be remodeled, and more accurate predictions could likely be obtained.

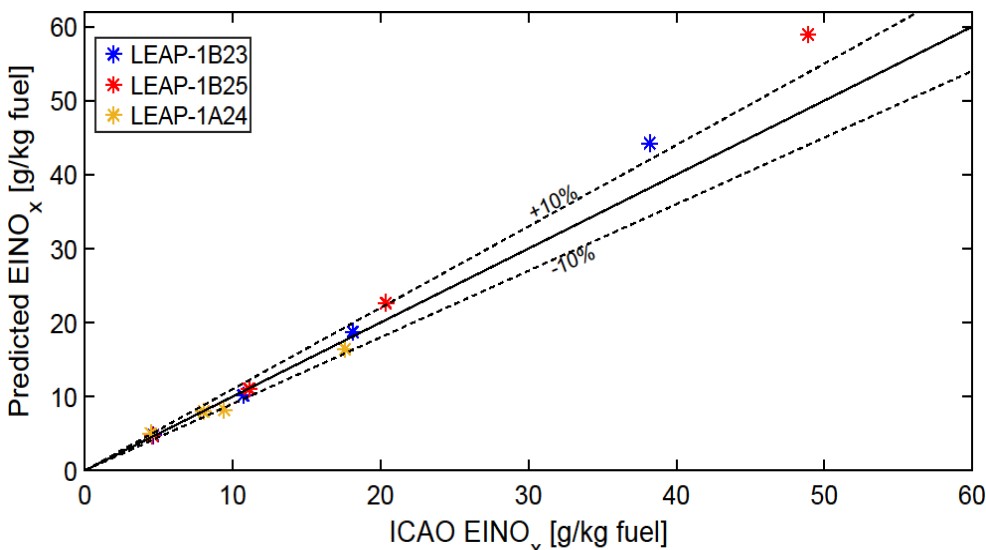

**Figure 6.** Comparison of the model−predicted EINO$_x$ (Equation (13)) with the ICAO−provided EINO$_x$ under SLS conditions for three LEAP models (total twelve points) for the TAPS−II combustor. The solid dashed line is used as a reference to measure the accuracy of the model, with the dashed lines representing a ±10% deviation.

The $EINO_{x,SLS}$ calculated from the above model was then corrected for changes in altitude and humidity to obtain $EINO_{x,cruise}$ using Equation (5), similar to the rich-burn correlations. The value of n was taken to be 0.2 [29]. m was obtained as −1.1598 from the curve-fitting results derived from A320neo simulations in PIANO-X in cruise mode under standard operating conditions, keeping *n* fixed. At three different cruise altitudes during the mission, the $EINO_{x,cruise}$ was obtained from PIANO, and the $EINO_{x,SLS}$ was calculated using Equation (13). The details of the simulation are given in the Appendix A.2.

Lean-Burn TAPS nvPM Model

For TAPS nvPM calculations, a similar approach as that used in the EINO$_x$ correlation was applied, with the pilot and pilot + main operation regimes treated separately. Based on the limited data available for the LEAP engines in the ICAO databank, three LEAP engine models were used for obtaining the $EI_{mass,SLS}$ and the ν correlations, and one engine model was used for validation. These include corrections for system losses during measurements,

as provided by the databank. For both operating regimes, exponential functions were implemented. The correlations are shown in Equations (14)–(16), respectively.

$$EI^p_{mass,SLS} = 0.0290 \cdot \exp(4.8961 \cdot t_1) + 0.9562 \cdot \exp(0.2439 \cdot t_1) \quad F_{engine} \leq 40\%$$
$$EI^{p+m}_{mass,SLS} = 0.0962 \cdot \exp(2.2046 \cdot t_2) + 1.4302 \cdot \exp(0.2152 \cdot t_2) \quad F_{engine} > 40\%$$
(14)

$$\nu^p = (7.316e + 13) \cdot \exp(0.68 \cdot t_1) \qquad F_{engine} \leq 40\%$$
$$\nu^{p+m} = (7.61e + 10) \cdot \exp(-0.1061 \cdot t_2) \quad F_{engine} > 40\%$$
(15)

$$t_1 = \frac{T_{t3} - 565.4}{100}$$
$$t_2 = \frac{T_{t3} - 817.1}{32.68}$$
(16)

Further, Equation (8) was used for calculating the cruise nvPM emissions, due to a lack of experimental and/or simulation data on offer to determine the exponents of the correction terms for a lean-burn combustor. Figure 7 shows the derived exponential curves, for both the operating regimes, along with the model prediction compared to the known ICAO data. When more data for other LEAP-X models are provided in the future in the ICAO databank, the correlations are expected to provide more accurate results, along with more points to validate them.

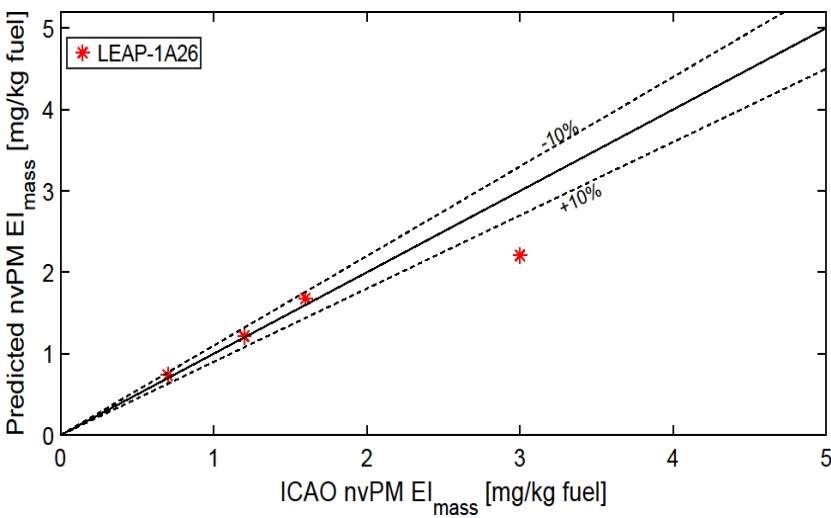

**Figure 7.** Comparison of the model−predicted nvPM $EI_{mass,SLS}$ (Equation (14)) with ICAO−provided nvPM under SLS conditions for one LEAP−X model (total four points) for the TAPS−II combustor. The solid dashed line is used as a reference to measure the accuracy of the model, with the dashed lines representing a ±10% deviation.

### 2.4. The Emission Inventory

An emission inventory is a 3D emission data set, containing, as a minimum, annual information about the fuel consumption, the flown distance and $NO_x$ emissions at a certain location in the atmosphere, which are described by the latitude, the longitude, and the pressure altitude, representing a specific atmospheric volume. The city pairs that form the emission inventory are shown in Figure 8, with an equal number of A320 flights in both regions. The geographical distribution of these flights is also shown. For these flights, the data from the OpenSky Network were generated with a time step of 30 s. For each city pair, first, the total great-circle distance was estimated. Then, each step of the modeling chain, as shown in Figure 2, was implemented, which gave the fuel consumption, the $NO_x$ and the nvPM emissions. For each city pair, one flight per day for one year was considered. Thus, the fleet consisted of 60 identical A320 aircraft, with the annual number of flights being 21,900. These data were then converted into a readable format that AirClim used as the emission inventory.

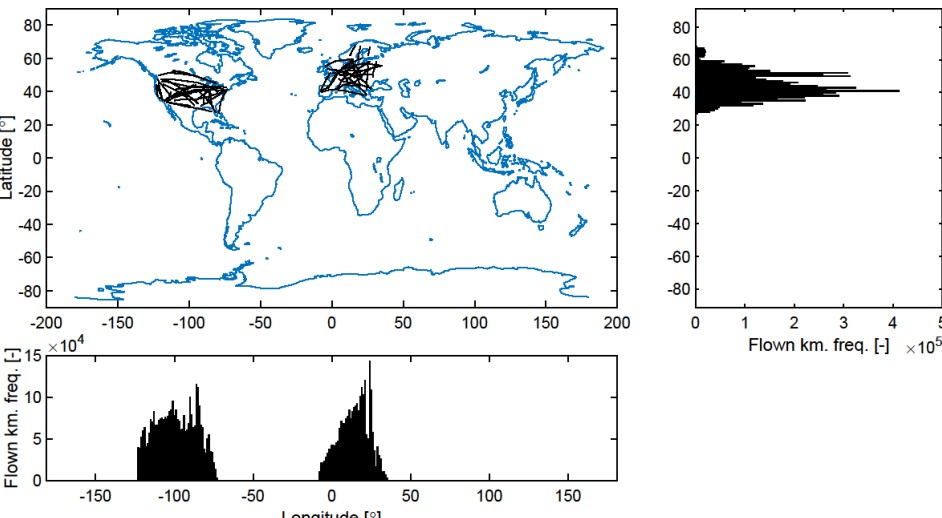

**Figure 8.** Flight planning consisting of the 60 city pairs. The annual global flown distance frequency distribution is also shown. The lines in the map indicate the flight routing between city pairs across North America and Europe. The bottom panel shows the flight frequency distribution along the longitudinal direction and the figure on the right panel shows the latitudinal distribution of the flight frequency.

### 2.5. Climate Assessment Model

After the creation of the emission inventory, the background modeling inputs in AirClim were determined. Some information about the past and (expected) future aviation fuel consumption, atmospheric chemical composition, and pre-calculated atmospheric input data obtained from sophisticated climate-chemistry models serve as further inputs into AirClim. AirClim is a non-linear response model, developed based on a detailed atmospheric chemistry process based on simulation results from a climate chemistry model (CCM), used to evaluate the climate impact of given aircraft engine technologies considering different scenarios. Non-$CO_2$ effects are complex as they have strong spatial and temporal resolutions. AirClim balances the level of details and the model fidelity, hence is a good candidate to evaluate aviation's climate impact. There are several climate metrics, e.g., the widely used global warming potential (GWP). The selection of the metric depends on the specific question to be answered. Megill (2023) [37] analyzed different metrics based on six criteria, and indicated that average temperature response (ATR) is the proper metric considering the stability and the direct indication of the global warming effect, and is a suitable metric as far as the comparison of aviation technology is concerned. Furthermore, the time horizon of 100 years balances the long-term and short-term effects.

Essentially, the aviation fuel consumption scenario and the atmospheric $CO_2$ and $CH_4$ concentration scenarios serve as the main background modeling inputs in AirClim. In this research, the CurTec scenario taken from Grewe et al. (2021) [15] was considered as the aviation fuel consumption scenario. Figure 9 shows the trend for the same. The scenario is described up to 2100 in the original file, whereas the trend beyond 2100 (dashed line) is assumed to be based on the annual fuel change rate in the year 2100.

For the atmospheric $CO_2$ and $CH_4$ concentration scenarios, the IPCC SSP2-4.5 scenario is considered [38]. This is the so-called "Middle of the Road" scenario, where the pace of sustainable development is relatively slow, and global resource and energy usage rates decline. Figure 10 shows the trends of the atmospheric concentration of the two species.

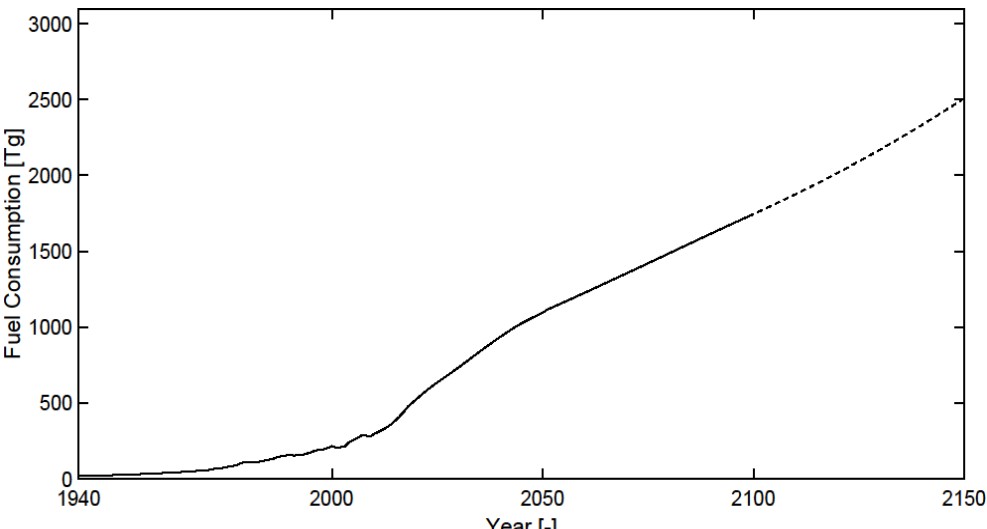

**Figure 9.** CurTec aviation fuel scenario [15]. The dashed-line-indicated trend beyond 2100 assumes that the annual fuel change remains the same as in 2100.

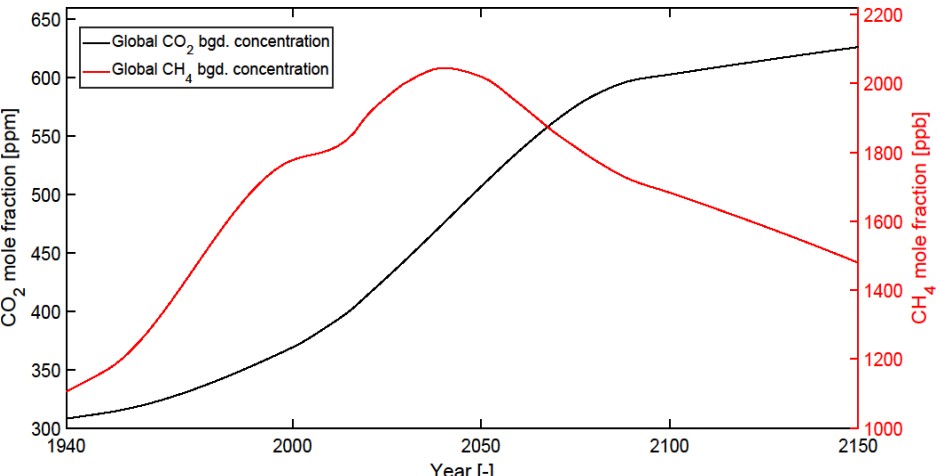

**Figure 10.** Background atmospheric $CO_2$ (in black) and $CH_4$ (in red) concentration trends for the IPCC SSP2-4.5 scenario [38] The units for the $CO_2$ trends are in parts-per-million (ppm), whereas for the $CH_4$ they are in parts-per-billion (ppb).

The complete methodology was repeated for each change in the OPR and TIT of the baseline engine. The OPR was varied from 25 to 40 in steps of 5. The TIT was varied from 1450 K to 1800 K. At the lower end of the TIT range, the resolution was kept smaller, at 25 K, to capture the finer trends of fuel consumption and $NO_x$ emissions. For other cases, the resolution was kept at 50 K. Further, the TIT was varied at the baseline OPR, and vice-versa. A total of 55 configurations were modeled in addition to the baseline engine. For each configuration, the cruise nvPM particle number emissions of the fleet were estimated using the nvPM emission model. Accordingly, the changes in contrail-induced RF ($RF_{contr.}$), i.e., $\Delta RF^{contr}$ were estimated relative to the baseline configuration using the correlation given in Grewe et al. (2021) [15], shown in Equation (17), where $\Delta pn$ is the relative change in particle number emissions of nvPM. It is to be noted that the correlation is only valid for $\Delta pn \geq 0.1$. This factor was then used in AirClim to determine the impact on the $RF_{contr.}$ and its subsequent temperature change ($\Delta T_{s.contr}$).

$$\Delta RF^{contr.} = \frac{arctan(1.9\Delta pn^{0.74})}{arctan(1.9)} \tag{17}$$

The climate simulations start in 2015 and end in 2150. The $ATR_{100}$ is measured from the year 2035, the year when technological options are representative. Thus, this is the year corresponding to which the background scenario is scaled according to the total fuel consumption from the various emission inventories. Essentially, the emission inventory represents the global distribution of emissions and flow distances for this year, and the temporal development of these is represented by the CurTec scenario shown in Figure 9.

### 2.6. Modelling Uncertainties

Since the analysis is highly multi-disciplinary in nature, there are uncertainties that can arise due to uncertainties within different sub-models. Regarding flight profiles, we consider only 60 city pairs with A320 being operational, which is a small subset of global aviation. One source of uncertainty is the factors/variables that have been taken to be constant but can change in practice, e.g., the payload. Another source of uncertainty in the modeling chain is the $NO_x$ and nvPM emission models. Even though the emissions have been validated using the four ICAO engine test points, the behavior of emissions between these points is interpolated. This may lead to some uncertainties in the emission predictions. In addition, the emissions are also affected due to changes in operating conditions, such as cruise altitude (considered fixed in this research), the engine power levels, and atmospheric humidity change. Kroon (2022) [39] analyzed the uncertainty of the $NO_x$ emission index subject to different sources, e.g., engine aging, which will be applied to the current research for uncertainty analysis. Regarding the climate model, using a different climate chemistry model or cloud scheme in AirClim (within the precalculated atmospheric input data [5,28]) may lead to deviations in the $ATR_{100}$. In addition, there are uncertainties related to the lifetime, climate sensitivity parameter and RF for the emission species [5,28].

## 3. Results and Discussion

This section discusses the results of the research. The key results that are discussed are the variations in fuel consumption, $NO_x$ emissions, nvPM particle number and the climate response in $ATR_{100}$. Together, they are called the resultant parameters in this research.

The results are presented in two parts. First, the values of the resultant parameters are discussed for the baseline engine (Section 3.1). They serve as the reference for the results of other engine configurations, which are presented next (Section 3.2). These are in the form of contour plots against the variation of the OPR (X-axis) and the TIT (Y-axis). The values of the resultant parameters are normalized with respect to the corresponding values for the baseline engine configuration. Further, the configurations with minimum fuel burn at each OPR are taken, and a comparison of the temporal development of various species' responses as well as of the $ATR_{100}$ is made.

### 3.1. Resultant Parameters for the Baseline Engine Configuration

Table 1 shows the resultant parameters for the baseline configuration. These represent the results at the fleet level, i.e., the resultant parameters across the 60 city pairs. The mean $EINO_x$ and nvPM $EI_{num.}$ are also shown. The nvPM particle number serves as the reference against which particle numbers for other configurations are compared, and Equation (17) is applied to estimate the $RF_{contr.}$ reduction. The $ATR_{100}$ measured from 2035 up to 2134 is 0.085 mK. This is the sum of the response of the six principal species calculated in AirClim: $CO_2$, $H_2O$, contrails, $NO_x$-induced $O_3$ enhancement, $NO_x$-induced $CH_4$ depletion and the subsequent PMO effects.

**Table 1.** Resultant parameters for the baseline CFM56. Fuel consumption and emissions are based on 60 city pairs.

| Config. | Fuel Consumption | $NO_x$ Emissions | Mean $EINO_x$ | nvPM No. ($\times 10^{20}$) | Mean $EI_{num.}$ ($\times 10^{14}$) | $ATR_{100}$ |
| --- | --- | --- | --- | --- | --- | --- |
| | (t(Fuel)) | (t($NO_2$)) | (g($NO_2$)/kg Fuel) | (#(nvPM)) | (#(nvPM)/kg Fuel) | (mK) |
| Baseline (A320 and CFM56) | 279.28 | 3.11 | 11.1 | 1.40 | 5.01 | 0.085 |

*3.2. Parametric Results*

3.2.1. Rich Burn Combustor Configurations

Variation in Fuel Consumption

Figure 11 shows the cruise fuel consumption trend normalized to the baseline configuration value. While fuel consumption reduces with increasing OPR, the dependency on TIT is not straightforward. For very low TITs (e.g., TIT = 1450 K), the engine specific thrust reduces, leading to a relatively larger core and increased fuel consumption with respect to baseline. Conversely, for very high TITs (e.g., TIT = 1800 K), there are more core nozzle losses, which again leads to higher fuel consumption with respect to baseline. This distinction becomes more prominent at higher OPRs. Hence, at every OPR, a TIT exists at which the fuel consumption is the lowest, and this TIT is higher for a higher OPR.

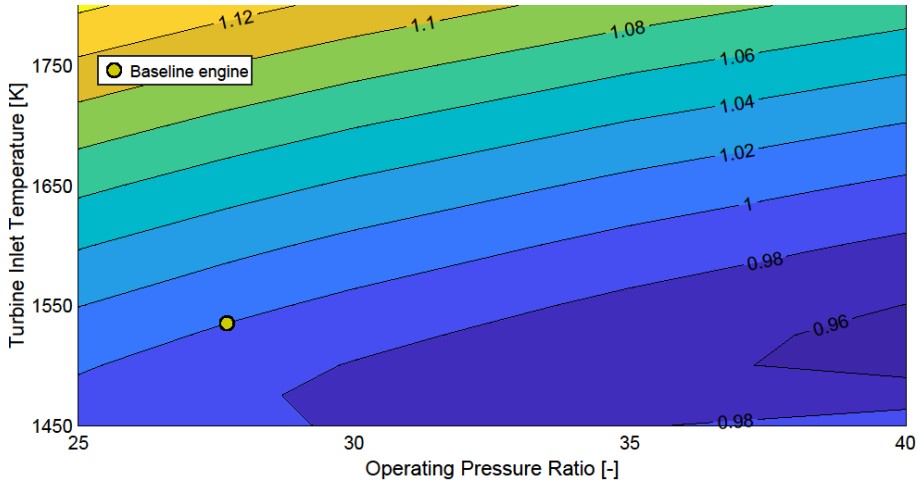

**Figure 11.** Cruise fuel consumption of the fleet with varying OPR and TIT for the rich-burn combustor configurations, normalized to the baseline (yellow circle), which is 279.28 t(fuel) (Table 1).

Variation of Emissions

Figure 12 shows the cruise $NO_x$ trend, normalized with the corresponding baseline engine $NO_x$ (3.11 t($NO_2$)). Aviation $NO_x$ production is mainly influenced by the pressure and temperature during combustion, which increases with OPR. Similarly, a higher TIT implies that the temperature in the combustor is still high, leading to higher $NO_x$ emissions with respect to baseline. Compared to Figure 11, it is observed that the respective reduction and increase in fuel consumption and $NO_x$ with changing design parameters are not similar. While fuel consumption typically reduces by less than 10% with increasing OPR, $NO_x$ can increase by 30 to 50%, depending on the TIT at which these trends are observed.

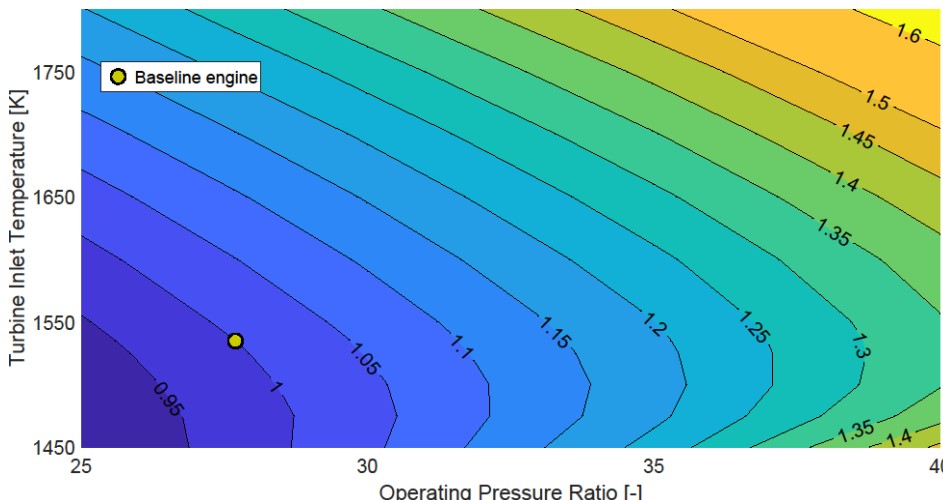

**Figure 12.** Cruise NO$_x$ emissions of the fleet with varying OPR and TIT for the rich-burn combustor configurations, normalized to the baseline (yellow circle), which is 3.11 t(NO$_2$) (Table 1).

The trend for the nvPM particles is shown in Figure 13. As the combustor inlet pressure and temperature increase with OPR, it makes conditions favorable for nvPM formation [40,41]. A high degree of non-linearity in nvPM formation with respect to increasing OPR is observed, with nvPM formation rapidly rising to be close to (and even greater at higher TITs) three times the baseline value at OPR = 40. In addition, an increasing TIT leads to greater nvPM formation due to the increased temperatures and fuel/air ratio within the combustor. At low OPRs (e.g., OPR = 25 and 30), nvPM formation exhibits mild sensitivity to TIT changes, and vice versa. Overall, the non-linear behavior of nvPM formation is mainly influenced by the formation of radical precursors (Warnatz et al., 2012) and changes in fuel consumption with design parameters, along with some degree of particle coagulation at high temperatures [23].

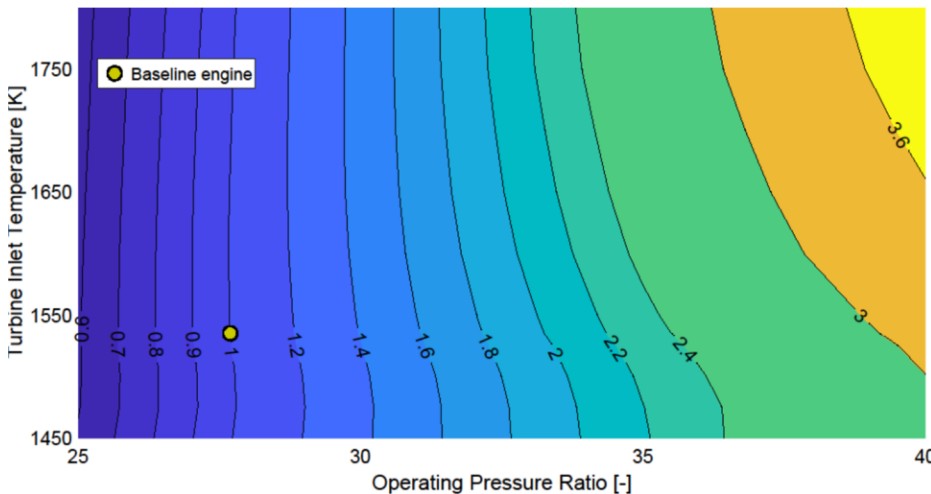

**Figure 13.** Cruise nvPM number of the fleet with varying OPR and TIT for the rich-burn combustor configurations, normalized to the baseline (yellow circle), which is $1.4 \times 10^{20}$ (#nvPM) (Table 1).

Variation of ATR$_{100}$

The ATR100 trend for the rich-burn combustor is shown in Figure 14. The ATR100 increases with OPR, with the trend being similar to the non-CO$_2$ emission trends shown in Figures 12 and 13. At low OPRs (e.g., OPR = 25 and 30), the increase in the TIT leads to smaller changes in the ATR100. As the OPR increases, the sensitivity of the ATR100 with respect to TIT also increases, as smaller increments in the TIT lead to large ATR100 changes.

This effect is more pronounced at high OPR and TIT configurations—see the top right area of the figure.

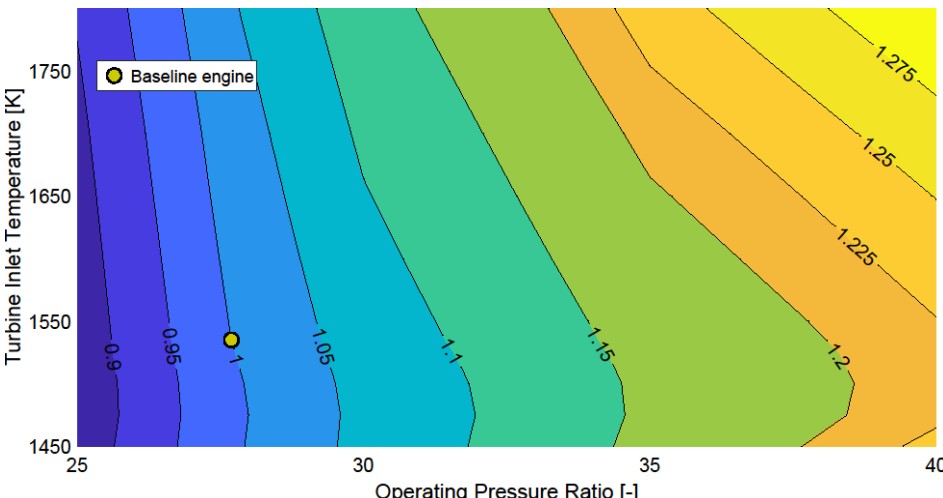

**Figure 14.** $ATR_{100}$ with varying OPR and TIT for the rich-burn combustor configurations, normalized to the baseline (yellow circle), which is 0.085 mK (Table 1).

### 3.2.2. Lean-Burn Combustor Configurations
Variation of Emissions

Figure 15 shows the cruise $NO_x$ trend for the TAPS-II configurations. Compared with Figure 12, the emissions' magnitude has reduced drastically, while showing similar trendlines. This reduction is larger at lower OPRs (e.g., OPR = 25 and 30), in the range of 30 to 40%, where low combustor pressure and temperature synergize well with lean combustion. The $NO_x$ emissions increase with the OPR and TIT, and the reduction with respect to the rich-burn configuration is only about 20% for OPR = 40. Further, a degree of non-linearity is also observed, around higher TIT values, indicating that $NO_x$ emissions can change rapidly for the TAPS-II combustor at higher design parameters, in contrast with the trends in Figure 12, where the contour lines are "smoother", indicating more monotonous trends.

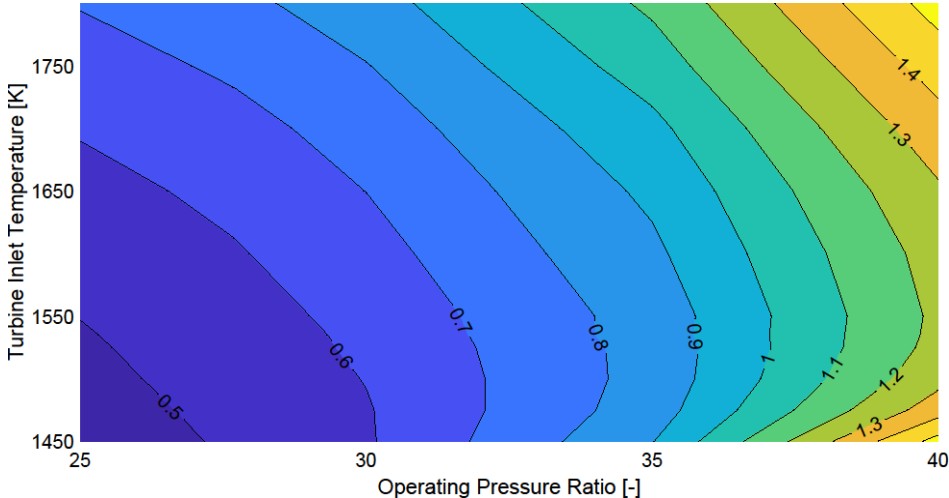

**Figure 15.** Cruise $NO_x$ emissions of the fleet with varying OPR and TIT for the TAPS-II combustor configurations, normalized to the baseline, which is 3.11 t($NO_2$) (Table 1).

The nvPM number trend for the TAPS-II configurations is shown in Figure 16. The TAPS-II combustor reduces the $EI_{num.,cruise}$ of nvPM by approximately four orders of

magnitude (1014 for the rich-burn combustor vs. 1010 for the TAPS-II) at cruise. A similar reduction is also observed in the data from the ICAO databank for engines with similar thrust levels. The lean combustion regime of the TAPS-II heavily influences the nvPM production, resulting in a drastic reduction in the particle numbers. The sensitivity of the nvPM number with respect to TIT has also increased, with it being a far more influential factor, especially at lower OPRs. Overall, with increasing OPR and TIT, the nvPM emissions increase, with the total emissions lower compared to the corresponding rich-burn configurations.

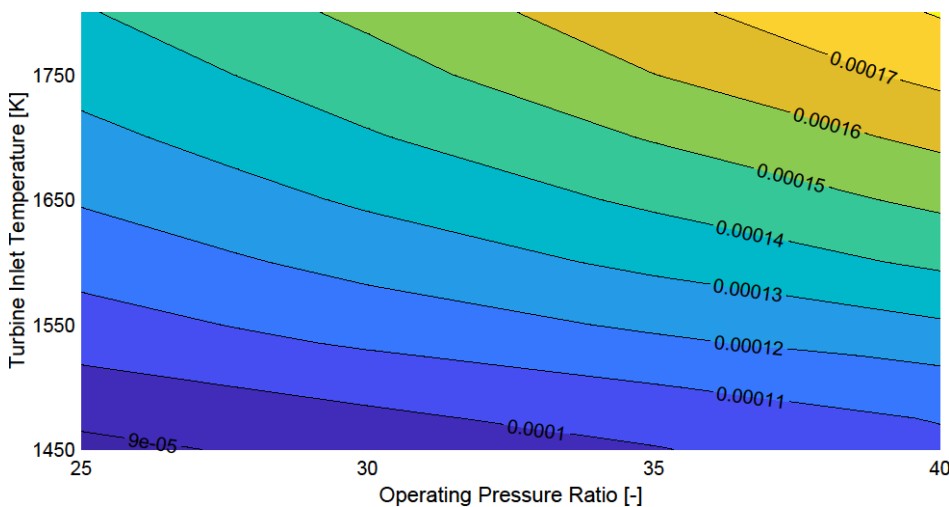

**Figure 16.** Cruise nvPM number of the fleet with varying OPR and TIT for the TAPS-II combustor configurations, normalized to the baseline, which is $1.4 \times 10^{20}$ (#nvPM) (Table 1).

Variation of $ATR_{100}$

The TAPS-II configuration ATR100 trend is shown in Figure 17. Relative to the corresponding rich-burn configuration, the ATR100 was reduced by approximately 50 to 60% for low OPRs (i.e., OPR = 25 and 30), and close to 70% for the higher OPRs (i.e., OPR = 35 and 40). Furthermore, with the change in the OPR, the change in the ATR100 was smaller, only about 10 to 12% from OPR = 25 to 40, compared to the rich-burn trend, where the same can be seen to be closer to 40%. In addition, the trend closely resembles the lean-burn $NO_x$ emission trend, shown in Figure 15.

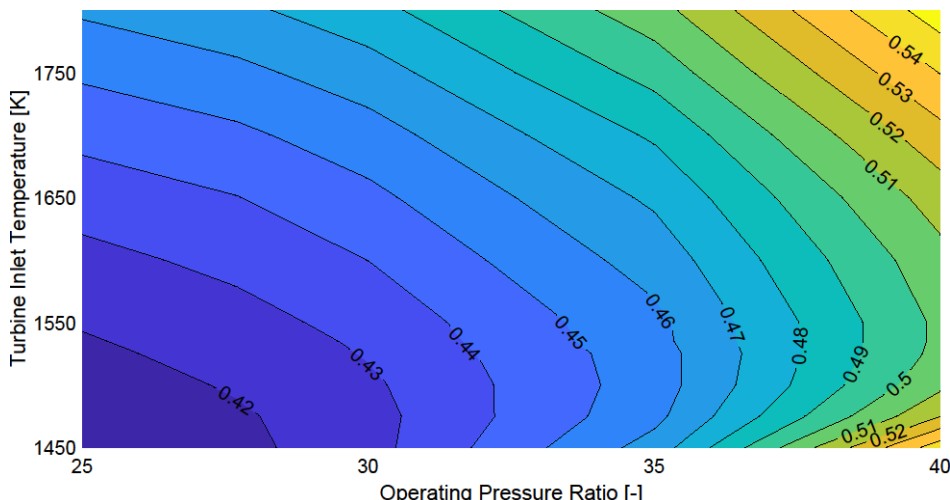

**Figure 17.** $ATR_{100}$ with varying OPR and TIT for the TAPS-II combustor configurations, normalized to the baseline, which is 0.085 mK (Table 1).

### 3.3. Climate Impact Analysis for Engines with a Minimal Fuel Burn

Irrespective of the combustion technology, the ATR100 trends are largely aligned with those of the two main non-$CO_2$ emissions discussed. To analyze the species' impact and their relative contribution, the temporal evolutions of the temperature response of the different species were analyzed. This enables the identification of species most critical for the total climate impact. In addition, the species' contributions to the total $ATR_{100}$ were also investigated, considering configurations with minimum fuel consumption at each OPR. The trend for the temperature response due to the $CO_2$ emissions is shown in Figure 18. The trends for both combustor technologies are identical, as there is no difference in the fuel consumption modeled between the two technologies. Throughout the simulation, the $CO_2$ response rose slowly, reaching 0.025 mK for the baseline engine in 2150, relative to 2015. This is a result of the well-mixed nature of $CO_2$, leading to a larger response time, on the order of decades to centuries. It can also be concluded from Figure 18 that the reduction obtained in the $CO_2$ response decreased as the OPR increased, with the lines for OPR = 35 and 40 placed closer to each other compared to OPR = 25 and 30.

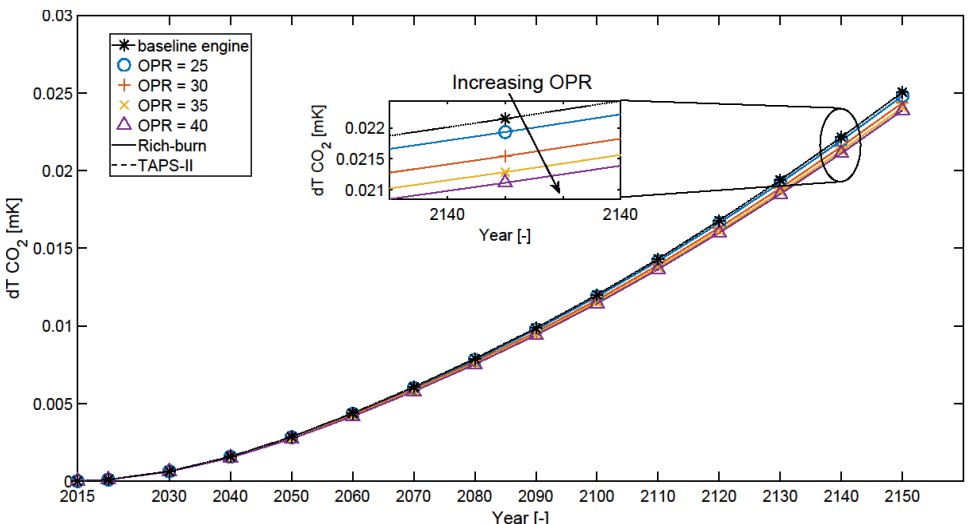

**Figure 18.** Temporal evolution of the temperature response due to $CO_2$ emissions (dT $CO_2$) for baseline engine (dotted-asterisk line) and the OPR range from 25 to 40. The trends are shown for both combustor technologies (in this case, they result in identical trendlines). The response due to the different configurations is also shown for the year 2140.

The trends for the $NO_x$ temperature response are shown in Figure 19. Not only was the $NO_x$ temperature response higher than the $CO_2$ temperature response, but the deviation from the baseline was also larger, with clear distinctions very early during the simulation. The difference between the two combustion technologies was much larger for lower OPRs, in line with the observations in Figures 12 and 15. It was also observed that the $NO_x$ response for OPRs 30 and 35 for the TAPS-II combustor was still lower than the baseline configuration, which highlights its potential use in reducing the impact of $NO_x$-related effects in engines operating at higher OPRs.

The contrail trends are shown in Figure 20. Effectively, with increasing OPR, the contrail response increased, due to the increased nvPM acting as a contrail precursor. While the TAPS-II combustor reduced the contrail effects relative to the rich-burn combustor due to the large reduction in nvPM, a threshold exists below which these effects diminished. At very low nvPM numbers, the particles were mixed in and formed droplets, which resulted in an approximately constant, low ice crystal number equaling the cloud condensation nuclei. However, there was some condensation of heavier volatile particles in the exhaust plume, e.g., sulfates, which contribute to contrail formation. This uncertainty in ice crystal

formation is defined by the $\Delta pn \geq 0.1$ limit in Equation (17), resulting in a constant response for the TAPS-II configurations with changing OPR, at 0.037 mK in 2150 relative to 2015.

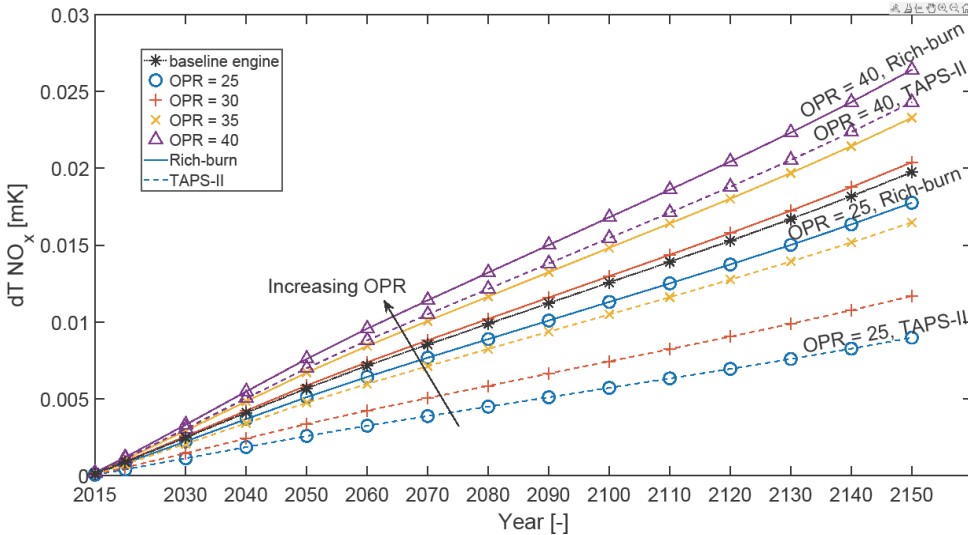

**Figure 19.** Temporal evolution of the temperature response due to $NO_x$ emissions (enhancement effects due to $O_3$ production and diminishing effects due to $CH_4$ and PMO) for baseline engine (dotted asterisk line) and the OPR range from 25 to 40. Solid lines represent rich-burn configurations, whereas dashed lines represent TAPS-II configurations.

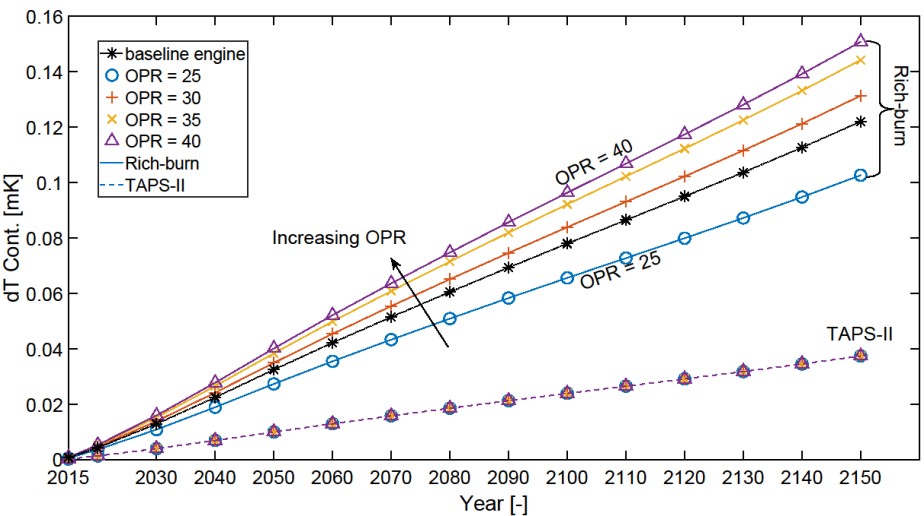

**Figure 20.** Temporal evolution of the temperature response due to contrail effects (dT Cont.) for baseline engine (dotted asterisk line) and the OPR range from 25 to 40. Solid lines represent rich-burn configurations, whereas dashed lines represent TAPS-II configurations.

The $NO_x$ and contrail responses combined made a large contribution to the total non-$CO_2$ response, as shown in Figure 21. There was a small contribution of $H_2O$ impact, although this was negligible. The non-$CO_2$ impact trendline showed a much steeper slope throughout the simulation compared to the $CO_2$ impact trendline, with the baseline engine configuration reaching up to 0.144 mK, in the year 2150, compared to the reference in 2015. For the TAPS-II combustor, the non-$CO_2$ response was lower, highlighting the effect of lean combustion on the nvPM formation and $NO_x$ emissions, which subsequently reduced the contrail and $NO_x$ effects.

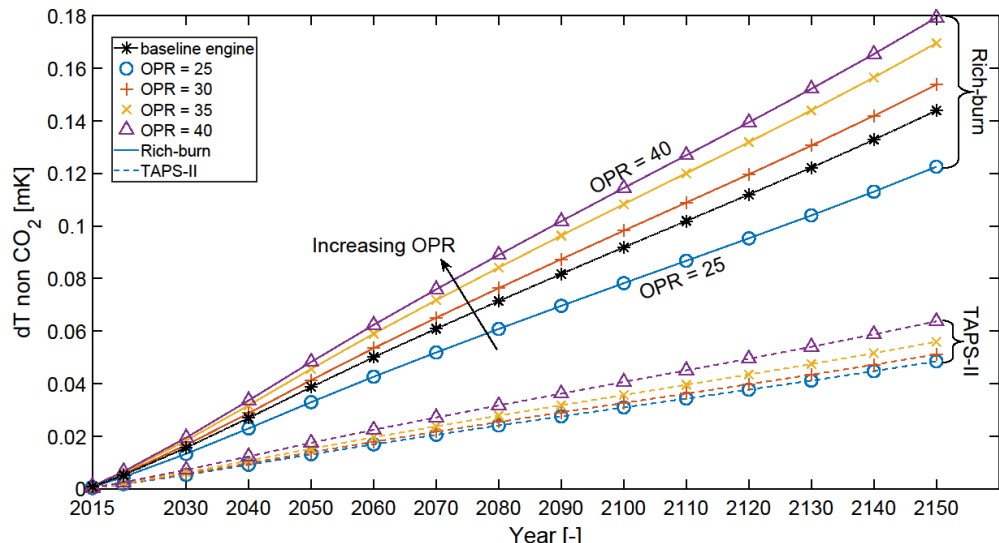

**Figure 21.** Temporal evolution of the temperature response due to the non-CO$_2$ emissions (dT NO$_x$, H$_2$O, and contrails) for baseline engine (dotted-asterisk line) and the OPR range from 25 to 40. Solid lines represent rich-burn configurations, whereas dashed lines represent TAPS-II configurations.

The total temperature response is shown in Figure 22, largely resembling the non-CO$_2$ impact trendline, due to the higher contribution and magnitude of these effects. It can be observed that the contrail response contributes the largest to the total climate impact. In 2150, the contribution of the non-CO$_2$ effects ranges typically from 83 to 88% for the rich-burn combustor and around 65–70% for the TAPS-II combustor, highlighting the difference in the sensitivities of the total response with respect to changing OPR between the rich-burn and TAPS-II combustors. Overall, the TAPS-II combustor shows a favorable performance in terms of climate response compared to the rich-burn combustor with increasing OPRs.

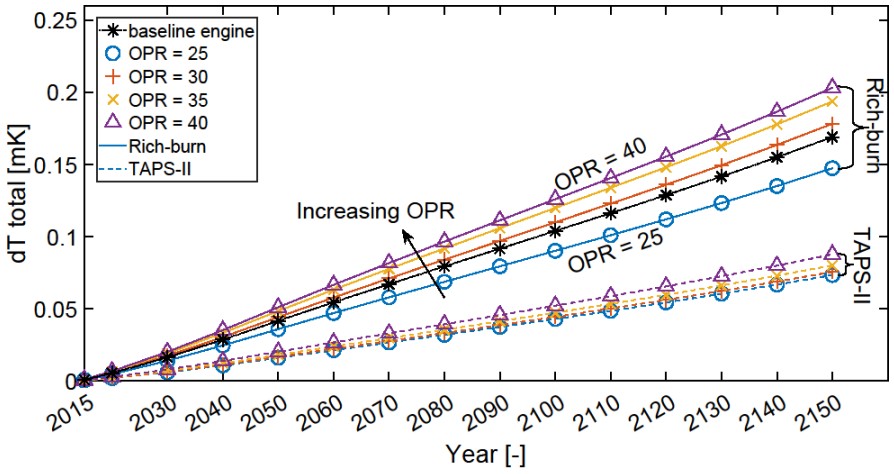

**Figure 22.** Temporal evolution of the total temperature response for baseline engine (dashed asterisk line) and an OPR ranging from 25 to 40. Solid lines represent rich-burn configurations, whereas dashed lines represent TAPS-II configurations.

To highlight the species' contributions over the 100-year horizon, Figure 23 shows the ATR100 for these configurations. Comparisons were made for the species with each other, as well as for the two combustor technologies. The total response increased with increasing OPR, corresponding to the trend in Figure 22, for both the rich-burn and TAPS-II configurations. Relative to the rich-burn combustor, the contrail climate impact reduced to the threshold value for the TAPS-II with increasing OPR, and subsequently, the relative

contribution decreased as well, with the contribution of $NO_x$ impact starting to increase. The contribution of the non-$CO_2$ effects ranged from approximately 85 to 92% for the rich-burn combustor and around 70–80% for the TAPS-II combustor, highlighting the significance of non-$CO_2$ emissions and their effects.

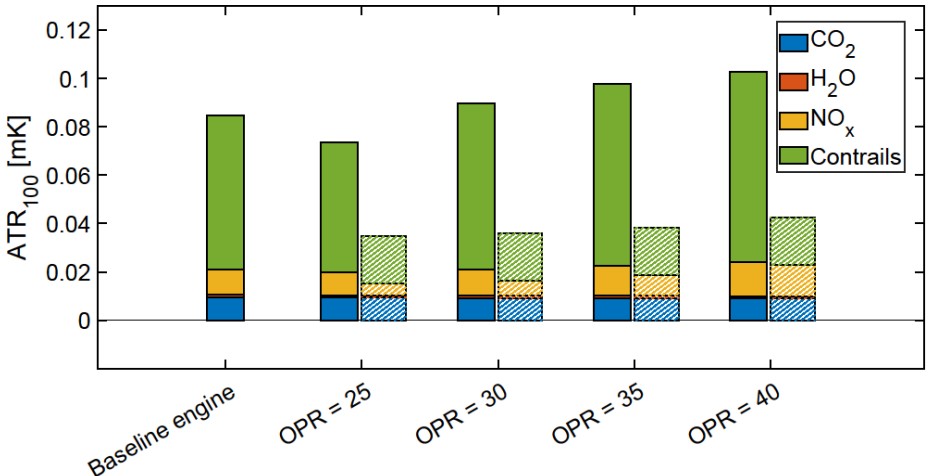

**Figure 23.** Comparison of the $ATR_{100}$ for each of the six species for the lowest fuel consumption configurations with changing OPR. The trends for the rich burn combustor are shown in solid, whereas those for the TAPS−II combustor are shown as hatched.

To quantitatively analyze the climate sensitivities with respect to engine design parameters, a parameter called the engine design species climate sensitivity has been introduced, which is given in Equation (18):

$$(\gamma_{spec})_h = \left( \frac{\left( \frac{\Delta ATR_{spec.}}{ATR_{baseline}} \right)}{\left( \frac{\Delta DesignParam.}{BaselineParam} \right)} \right)_h \tag{18}$$

h is the time horizon, i.e., 100 years. This parameter can be used to analyze the total and the individual species' sensitivities with respect to engine design. Some exemplary results are given in Table 2, where the sensitivity of a few key species and the total impact with respect to the changing OPR are shown for the two combustion technologies. Similar sensitivities can be estimated for any range of parameters within the design space.

**Table 2.** $(\gamma_{spec})_{100}$ values for $CO_2$, $NO_x$, contrails, and total climate impact for two exemplary cases over a time horizon of 100 years for the two combustion technologies.

| Design Parameter | Combustor Technology | $(\gamma_{total})_{100}$ | $(\gamma CO_2)_{100}$ | $(\gamma NO_x)_{100}$ | $(\gamma_{Contrails})_{100}$ |
|---|---|---|---|---|---|
| OPR (27.69 to 40 at TIT = 1535 K) | Rich burn | 0.4964 | −0.0989 | 0.7645 | 0.5512 |
| OPR (27.69 to 40 at TIT = 1535 K) | TAPS-II | 0.4112 | −0.0989 | 2.7404 | 0 |

Some interesting insights can be gained from Table 2. The $\gamma_{CO2}$ indicates a reduction in the $CO_2$ response with increasing pressure ratio. $ATR_{100,NO_x}$ was the most sensitive to OPR increases, with the TAPS-II configuration showing much higher sensitivity. Further, in this comparison, the $\gamma_{Contrail}$ and $\gamma_{total}$ in the rich-burn configuration showed similar magnitudes of change, indicating that contrails are the critical factor in the total climate impact. The $\gamma_{contrails}$ values did not change in the TAPS-II case, due to the contrail RF reduction threshold.

In this research, we investigated how the engine core design and combustion technology development of a civil turbofan engine has an impact on global temperature change based on a subset of the global flight network for a short-to-medium-range aircraft. The

climate impacts of $CO_2$, $NO_x$, $H_2O$, and contrails were included. The OPR and the TIT were systematically changed within a design space to create a matrix of engine configurations for two different combustion technologies, the rich-burn combustor and the lean-burn TAPS-II combustor. Corresponding to each of these configurations, the $ATR_{100}$ climate metric was calculated and an exhaustive comparison was made. In this research, other aspects of engine integration such as engine weight, size, and installation drag penalties were not considered.

The climate impact of emissions adds a critical dimension as far as engine designs (current and future) are concerned and is currently not included in the engine design stage. In addition, this research also aims to raise general awareness, particularly about non-$CO_2$ emissions and their climate effects in aviation. The general discourse tends to focus on $CO_2$ emissions and affinity towards flying; however, non-$CO_2$ emissions are responsible for a larger contribution towards the climate impact of aviation, as these emissions tend to stay for longer in the atmosphere due to a lack of washout processes. With an increase in passenger traffic as predicted by ICAO, it becomes important that the key issues are focused upon, and policies be formulated in accordance.

## 4. Conclusions

The results demonstrate that for short-to-medium-range aircraft such as the A320, increasing the engine OPR and TIT will lead to higher $NO_x$ and nvPM emissions. Also, the reduction in $CO_2$ impact due to reduced fuel consumption is not enough to compensate for such negative effects. With increases in the OPR from 25 to 40, the total $ATR_{100}$ increases by 40%, mainly dominated by the increase in the $NO_x$ and contrails' effects by 49% and 47%, respectively, for rich-burn combustion. The TAPS-II combustor reduces the total $ATR_{100}$ by 52% and 58% for OPR = 25 and 40, respectively. This is due to the contrails' effects being reduced by 63% and 75%, and the $NO_x$ effects being reduced by 49% and 8% for OPR = 25 and OPR = 40, respectively. These results provide novel insights into designing turbofans with the climate impact perspective in consideration, although these are dependent on the chosen flight profiles.

The uncertainties in the results may be due to emission modeling, flight profiles and the climate model implemented. Some of these uncertainties can be reduced to a large extent by using an extensive global flight inventory, a detailed combustion model and integrating Monte-Carlo simulations within the modeling chain.

**Author Contributions:** Conceptualization, F.Y.; methodology, F.Y. and H.S.S.; validation, H.S.S. and F.Y.; software, H.S.S.; formal analysis, H.S.S.; investigation, H.S.S.; writing—original draft preparation, F.Y. and H.S.S.; writing—review and editing, F.Y., A.G.R., V.G. and H.S.S.; supervision, F.Y., A.G.R. and V.G.; funding acquisition, F.Y. All authors have read and agreed to the published version of the manuscript.

**Funding:** This research was funded by the Dutch Research Council (NWO) with the grant number 17367 and the Horizon Europe with the grant number 101056863.

**Data Availability Statement:** The dataset produced in this study is available on the 4TU.ResearchData repository with the DOI: 10.4121/edf74400-173c-4efb-9c68-43ee1a090bdd.

**Conflicts of Interest:** The authors declare no conflict of interest.

## Abbreviations

| | |
|---|---|
| ACARE | Advisory Council for Aviation Research and Innovation in Europe |
| ATR | Average Temperature Response |
| BPR | Bypass Ratio |
| CCM | Climate Chemistry Model |
| CORSIA | Carbon Offsetting and Reduction Scheme for International Aviation |
| EI | Emission Index |

| ERF | Effective Radiative Forcing |
| FAR | Fuel-to-Air ratio |
| GSP | Gas Turbine Simulation Program |
| GWP | Global Warming Potential |
| ICAO | International Civil Aviation Organization |
| LB | Lean Burn |
| LEAP | Leading Edge Aviation Propulsion |
| LTO | Landing and Take-off Cycle |
| nvPM | Non-volatile Particulate Matter |
| OEW | Operating Empty Weight |
| OPR | Operating Pressure Ratio |
| PMO | Primary-mode Ozone |
| RF | Radiative Forcing |
| RQL | Rich-Burn Quick-Quench Lean-burn |
| SLS | Static Sea-Level |
| TAPS | Twin Annular Premixed Swirler |
| TIT | Turbine Inlet Temperature |
| TSFC | Thrust-Specific Fuel Consumption |

**Symbols**

| $\text{ATR}_{100}$ | Average Temperature Response over 100 years |
| $\text{ATR}_{baseline}$ | ATR for the baseline configuration |
| BaselineParam. | Baseline Engine Design Parameter |
| $d_{cruise}$ | Cruise Distance |
| $EI_{mass,cruise}$ | nvPM Cruise Mass Emission Index |
| $EI_{mass,SLS}$ | nvPM SLS Mass Emission Index |
| $EI^{p}_{mass,SLS}$ | nvPM SLS Mass Emission Index for *pilot* Operation |
| $EI^{p+m}_{mass,SLS}$ | nvPM SLS Mass Emission Index for *pilot + main* Operation |
| $EINO_{x,cruise}$ | Cruise $NO_x$ Emission Index |
| $EINO_{x,SLS}$ | SLS $NO_x$ Emission Index |
| $EINO^{p}_{x,SLS}$ | SLS $NO_x$ Emission Index for *pilot* Operation |
| $EINO^{p+m}_{x,SLS}$ | SLS $NO_x$ Emission Index for *pilot + main* Operation |
| $EI_{num,cruise}$ | nvPM Cruise Number Emission Index |
| $EI_{species}$ | Species Emission Index |
| $FAR_{cruise}$ | Cruise FAR |
| $FAR_{SLS}$ | SLS FAR |
| $F_{avg.aircraft}$ | Average Aircraft Cruise Thrust |
| $F_{avg.engine}$ | Average Engine Cruise Thrust |
| $F_{engine}$ | Engine Thrust Level |
| $(\gamma_{spec.})_{h}$ | Engine Design Species Climate Sensitivity over a Time Horizon of *h* Years |
| $h$ | Atmospheric Humidity |
| $h_{cruise}$ | Cruise Altitude |
| $L/D$ | Lift-to-Drag Ratio |
| $m$ | FAR-Term Exponent in P3–T3 Correlation |
| $n$ | Pressure-Term Exponent in P3–T3 Correlation |
| $P_{t3}$ | Combustor Inlet Pressure |
| $P_{t3,cruise}$ | Cruise Combustor Inlet Pressure |
| $P_{t3,SLS}$ | SLS Combustor Inlet Pressure |
| $RF_{contr.}$ | Contrail Radiative Forcing |
| $t_{cruise}$ | Cruise Time |
| $T_{t3}$ | Combustor Inlet Temperature |
| $\dot{w}_{a}$ | Intake Air Mass Flow Rate |
| $W_{app.payload}$ | Apparent Payload Weight |
| $W_{end}$ | Aircraft Weight at the End of Cruise |
| $W_{f,cruise}$ | Cruise Fuel Weight |
| $\dot{w}_{f,cruise}$ | Cruise Fuel Flow Rate |
| WNOx | $NO_x$ Emission Weight |

| | |
|---|---|
| $W_{species,cruise}$ | Weight of Emission Species |
| $Wstart$ | Aircraft Weight at the Beginning of Cruise |
| $\Delta ATR_{spec.}$ | Change in Species ATR |
| $\Delta DesignParam.$ | Change in Engine Design Parameter |
| $\Delta pn$ | Relative Change in nvPM Number Emissions |
| $\Delta RF^{contr}$ | Change in Contrail Radiative Forcing |
| $\Delta T_{s.contr}$ | Temperature Change due to Contrail-Induced Radiative Forcing |
| $\nu$ | nvPM Number-to-Mass Emission Index Ratio |
| $\nu^P$ | nvPM Number-to-Mass Emission Index Ratio for *pilot* Operation |
| $\nu^{p+m}$ | nvPM Number-to-Mass Emission Index Ratio for *pilot + main* Operation |

## Appendix A

*Appendix A.1. Validation Cases of the Aircraft Performance Model*

This section provides the details of the validation cases for the aircraft performance model. The input parameters to the performance model for these are shown in Table A1. The cruise altitude, $h_{cruise}$, for these cases was fixed at 35,000 ft. The TSFC data for different engines were obtained from open sources (https://www.janes.com/ (accessed on April 2023)) [42].

**Table A1.** Input parameters of the aircraft performance model used for validation.

| Aircraft | Cruise Mach No. | Cruise Distance | Cruise Fuel Weight | Operating Empty Weight | Apparent Payload Weight | Lift-to-Drag Ratio | Thrust-Specific Fuel Consumption |
|---|---|---|---|---|---|---|---|
| | M | $d_{cruise}$ | $W_{f,cruise}$ | OEW | $W_{app.payload}$ | L/D | TSFC |
| | (-) | (km) | (kg) | (kg) | (kg) | (-) | (kg/Ns) |
| A300-600R | 0.8 | 4587 | 27,852 | 89,813 | 32,519.23 | 15.51 | $1.637 \times 10^{-5}$ |
| A340-642 | 0.82 | 13,405 | 127,337 | 181,100 | 45,015.38 | 19 | $1.526 \times 10^{-5}$ |
| B767-300ERW | 0.82 | 10,619 | 59,168 | 93,032 | 30,636.48 | 18.31 | $1.632 \times 10^{-5}$ |

*Appendix A.2. PIANO Simulations of the A320neo Aircraft for TAPS-II Exponent Determination*

Table A2 shows the details of the PIANO-X simulation of the A320neo aircraft under design range and standard payload conditions. The $EINO_x$ from these simulations has been used to obtain the FAR exponent for the P3–T3 correlation, which is used to calculate the $NO_x$ emissions for the TAPS-II configurations.

**Table A2.** Cruise condition data for the A320neo in the design range and under standard payload conditions, obtained from PIANO-X.

| Cruise Operating Conditions | Cruise Distance (Km) | $W_{f,cruise}$ (kg) | $WNO_x$ (g) | $EINO_x$ (g/kg fuel) |
|---|---|---|---|---|
| $h_{cruise}$ = 34,000 ft Mach no. = 0.775 | 1025.1 | 2920.4 | 16.79 | 5.75 |
| $h_{cruise}$ = 36,000 ft Mach no. = 0.775 | 2906.41 | 7703.3 | 43.07 | 5.6 |
| $h_{cruise}$ = 38,000 ft Mach no. = 0.775 | 2110.6 | 5115 | 27.20 | 5.32 |

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
