# Peer review of "Effect of Engine Design Parameters on the Climate Impact of Aircraft: A Case Study Based on Short-Medium Range Mission"

_aerospace, doi:10.3390/aerospace10121004_

Round 1
Reviewer 1 Report
Comments and Suggestions for Authors
Introduction:
There is a good explanation of the impact of emissions on climate change. However, the possibility of optimizing the engine construction, with regard to emissions, is not sufficiently explained. It is mentioned in the abstract (sentence about OPR, BPR and TIT, line 10 and 11), but it is not explained well enough in the Introduction and in the section 2.2. Engine performance model
Methodology:
It is necessary to explain what AirClim and ATR100 are (softwares, programs, models or methods) and why exactly they are used? Are there any other software tools for the same purposes?
It is necessary to give references for the Equations (1) to (7) like a reference is given for the Equation (8).
It is necessary to explain what pilot and pilot+main operation are (line 265)
References:
Citation of reference in the text of the article and list of references at the end of the article is not in accordance with the instructions and template given by MDPI.
Author Response
The authors would like to thank you for the constructive feedback of the reviewer.
The paper has been revised to implement the comments. Please see the changes in the highlights of the manuscript.
A point-to-point reply is attached here

Reviewer 2 Report
Comments and Suggestions for Authors
The paper aims to evaluate the climate impact of turbofan engine design in terms of cycle development and combustion technologies over time, specifically it is the Average Temperature Response over 100 years.
The methodology relies on a complete model chain consisting of aircraft and engine models, emissions models for the prediction of NOx and non-volatile particulate matter, a state-of-the-art climate assessment model via the AirClim tool.
The analysis is carried out by varying two fundamental parameters for turbofan engines, say: the operating pressure ratio and the turbine inlet temperature. Therefore, depending on these two parameters the variation of fuel consumption, NOx emissions, nvPM particle number and the climate response in ATR100.
Results demonstrated that for short-to-medium-range aircraft such as the A320, increasing the engine OPR and TIT will lead to higher NOx and nvPM emissions. Also, the reduction in CO2 impact due to reduced fuel consumption is not enough to compensate such negative effects.
The paper is interesting, well presented and well written. However, in my opinion some details should be, clarified, deepened or modified.
I suggest changing the title, the work is a numerical prediction based on statistics.
If a software (PIANO) is already used to evaluate the average thrust, time and distance of aircrafts, what is the novelty of the in-house tool proposed by the authors?
References are too old.
I think that the addition of a nomenclature and abbreviation section is mandatory.
Row 136: “dcruise”, cruise should be a subscript.
Section 2.6: I suggest writing this section in bullet points.
The language style of the results section should be revised, some terms are repeated too often.
Rows 393, 432, 441, 470, 591: “w.r.t.” please write it in the extended form.
About Figure 7, at rows 508 to 511 the maning of the different symbols is reported, I think it is better to report a legend in the same figure, in this way the figure alone is self-explained. If the authors cannot modify the figure, I suggest writing the symbols meaning as bullet points.
The same consideration is valid for Figure 22 as well.
Finally, from what I understood for the evaluation of the emission only an emission index is considered. How is the emission index evaluated by changing of the operating conditions? I think a detailed description of this part is necessary.
Author Response
Thank you for the constructive feedback of the reviewer. We implemented the comment in the revision as highlighted.
A point-to-point reply is also attached here.

Round 2
Reviewer 1 Report
Comments and Suggestions for Authors
The revised paper is in accordance with the comments and suggestions.
Reviewer 2 Report
Comments and Suggestions for Authors
The authors answer to all of my questions.